# Implant-Prosthetic Restoration of a Patient with Osteogenesis Imperfecta: A Case Report

**DOI:** 10.3390/ijerph18084169

**Published:** 2021-04-15

**Authors:** Marcel Hanisch, Melanie Maus, Johannes Kleinheinz

**Affiliations:** Research Unit Rare Diseases with Orofacial Manifestations, Department of Cranio-Maxillofacial Surgery, Albert-Schweitzer-Campus 1, University Hospital Münster, Building W 30, D-48149 Münster, Germany; melanie.maus@ukmuenster.de (M.M.); johannes.kleinheinz@ukmuenster.de (J.K.)

**Keywords:** osteogenesis imperfecta, implant dentistry, oral health-related quality of life, rare diseases

## Abstract

Osteogenesis imperfecta describes a group of genetic disorders that result from a defect in collagen type I and range in severity from a subtle increase in fracture frequency to death in the perinatal period. Osteogenesis imperfecta is mostly caused by mutations in the COL1A1 (17q21.33) and COL1A2 (7q21.3) genes. There have only been a few case reports of implant-prosthetic treatment for patients with osteogenesis imperfecta. These reports indicated that implants and augmentation procedures can be implemented in such patients. However, for patients receiving additional antiresorptive therapy, cautious approaches should be chosen and the risk of drug-associated osteonecrosis should be considered. The aim of this article is to report on the implant-prosthetic treatment of a patient with type I osteogenesis imperfecta.

## 1. Introduction

Osteogenesis imperfecta (OI) describes a group of genetic disorders that result from a defect in collagen type I [1] and range in severity from a subtle increase in fracture frequency to death in the perinatal period [2,3]. The causes of OI primarily include mutations in the COL1A1 (17q21.33) and COL1A2 (7q21.3) genes [4]. In this regard, OI is divided into five clinically distinguishable types, with increased bone fragility and a tendency to develop spontaneous fractures being the predominant features of all types of OI [4]. Bisphosphonates (BPs) are the most widely used interventional modes of therapy due to their beneficial effects on bone mineral density [1]. Dental abnormalities in the form of dentinogenesis imperfecta are commonly observed in OI patients [5] but do not occur in all types of OI [4].

Orofacial manifestations often associated with OI include DI (Dentinogenesis imperfecta), posterior open bite (lateral open bite), class III dental and skeletal malocclusion, anterior and posterior crossbites and impacted teeth [6].

A classification system based on the severity of bone fragility, as indicated by clinical/radiological features, was proposed by Sillence et al. in 1979. Specifically, Sillence et al. proposed four phenotypic categories of OI: OI type I, nondeforming with blue sclera; OI type II, perinatally fatal OI; OI type III, progressively deforming OI; and OI type IV, moderately severe OI [7]. Other types of OI (V and above) have also been described based on specific phenotypic features and genetic findings. In Nosology and Classification of Genetic Skeletal Diseases, published in 2015, a phenotypic criterion was adopted to classify the first five types of OI (OI types I–V) [8].

The mildest form of OI is considered to be OI type I, in which collagen type I is of normal quality but reduced quantity [9]. To date, there have only been a few case reports of implant-prosthetic restorations in patients with OI [10,11].

It is known that immunosuppressed patients have a higher risk of implant loss [12,13,14,15]. The aim of this article is therefore to report on the implant-prosthetic restoration of a patient with OI type I.

## 2. Case Report

A 64-year-old otherwise healthy female patient presented for the first time in May 2019 for a specialty consultation concerning rare diseases with oral involvement. In this consultation, the patient reported a history of OI type I. Due to this underlying disease, she had experienced frequent fractures since childhood, most recently including a multiple-fragment fracture of the humerus that occurred during a fall at her home.

The patient had received alendronic acid p.o. for many years to treat the underlying disease. Two years prior to the consultation, this therapy was replaced with denosumab injections. Approximately six months prior to the consultation, the patient’s last teeth were removed alio loco under perioperative, prolonged antibiosis with plastic coverage. At the time of the consultation, she was fitted with removable maxillary and mandibular dentures. The insufficient support of the mandibular denture caused her discomfort and made it difficult for her to eat. Therefore, she wished to stabilize the prosthesis with implants.

Clinically, the patient presented with an edentulous maxilla and mandible without signs of os liber. The soft tissues were closed and inconspicuous, and there was no swelling or redness. In particular, the mandibular alveolar process showed pronounced vertical and horizontal atrophy (Figure 1). In addition, the patient exhibited a discoloration of the sclerae, which is characteristic of OI type I (Figure 2). Preoperative panoramic view, Figure 3.

Following a risk assessment, the patient was offered the insertion of two interforaminal implants to stabilize the prosthesis. Augmentation measures were not used due to the underlying disease and the patient’s history of drug therapy with alendronic acid and denosumab.

Under local anesthesia and perioperative prolonged antibiosis (clindamycin 600 mg 1-1-1) due to penicillin intolerance for seven days in case of known penicillin intolerance, beginning on the day of surgery), a crestal incision was made without relief incisions from regions 35 to 45. After bilateral exposure of the mental nerve, two narrow areas were identified in regions 32 and 44, in which the insertion of two tissue-level implants (Straumann Standard Plus^®^, Basel, Switzerland, SLActive 3.3 mm × 10 mm) was possible without the use of augmentation measures (Figure 4).

The patient’s postoperative wound healing was unremarkable, and the sutures were removed seven days after surgery. Furthermore, there were no occurrences of dehiscence or signs of drug-associated osteonecrosis. After a healing period of three months, uncovery was performed (Figure 5) so that prosthetic restoration could be carried out with locators (Figure 6) two weeks later.

Since the surgery, the patient has attended regular follow-ups for one year and exhibited no signs of drug-associated osteonecrosis or peri-implant infection (Figure 7 and Figure 8). The patient tested negative for bleeding on probing at both implants, with probing pocket depths (4-point measurements) of 2-1-3-2 at region 32 and 2-2-2-1 at region 42. Oral hygiene was very good. Radiological control was omitted given the patient’s good clinical parameters. The patient herself described a significantly improved prosthesis fit and an associated increase in quality of life.

## 3. Discussion

To date, only a few case reports have been published on implant-prosthetic restorations in patients with OI [10,16,17,18,19,20,21,22]. A Norwegian study reported a 95.7% survival rate of implants in patients with OI 25–135 months after prosthetic loading [22]. Successful augmentative procedures using autologous retromolar bone, corticospongous grafts of the iliac crest and synthetic substitutes have also been reported [10,11,16,17,18]. Thus, although the relevant data remain limited, the use of implants and augmentative measures in patients with OI seems to be safe and achievable in principle, without any major limitations.

As in the case report presented herein, antiresorptive therapy with BPs or denosumab often accompanies OI and must be taken into account. In particular, BPs represent the most widely used interventional therapy for OI due to their beneficial effects on bone mineral density [1]. The currently available literature indicates that children and young adults with OI and concomitant BP therapy are not at risk for drug-associated osteonecrosis [23,24,25,26]. However, drug-associated and implant-associated osteonecrosis was reported in a 75-year-old female patient with OI who had received BP therapy with alendronic acid for ten years [27]. Given that the patient in the present case was of a comparable age and had also been treated with alendronic acid for several years, we considered the risk of drug-associated osteonecrosis and discussed and weighed the potential consequences in detail with the patient.

The alternative to the surgical intervention would have been to leave the purely mucosa-supported denture in place. In principle, however, prior studies have shown a clear correlation between denture-related pressure points and the occurrence of osteonecrosis [28]. The reduction in mucosal stress associated with an implant-retained prosthesis can therefore reduce the risk of osteonecrosis caused by pressure points [28]. Consequently, restoration with a higher number of implants and a different prosthetic solution (e.g., a bar restoration) could have achieved further reductions in mucosal load in the present case. Although the literature has reported successful cases of augmentation and implant placement in patients with BP treatment histories and low-risk profiles [29], we decided against this strategy due to the significantly reduced bone supply of the patient and the concomitant need for prior augmentation. However, the risks of osteonecrosis remain and should not be underestimated [30]. Biomaterials may be a helpful option in the future for reduced bone supply [31,32].

## 4. Conclusions

Implants and augmentative measures can be implemented in patients with OI. For patients who have received additional antiresorptive therapy, cautious procedures should be chosen and the risk of drug-associated osteonecrosis should be considered.

## Figures and Tables

**Figure 1 ijerph-18-04169-f001:**
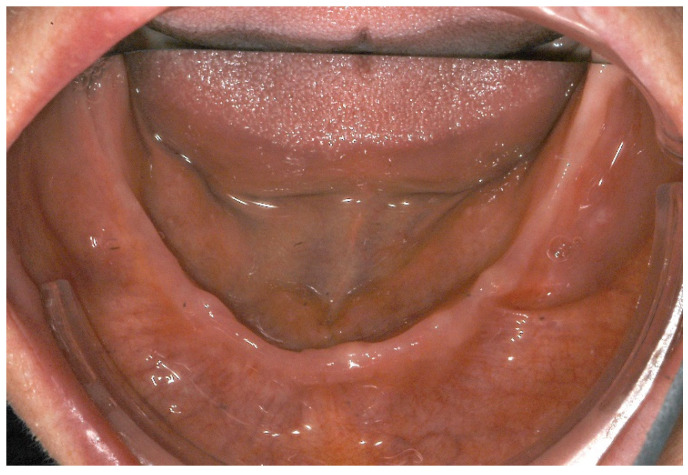
Intraoral situation with pronounced atrophy of the mandibular alveolar process.

**Figure 2 ijerph-18-04169-f002:**
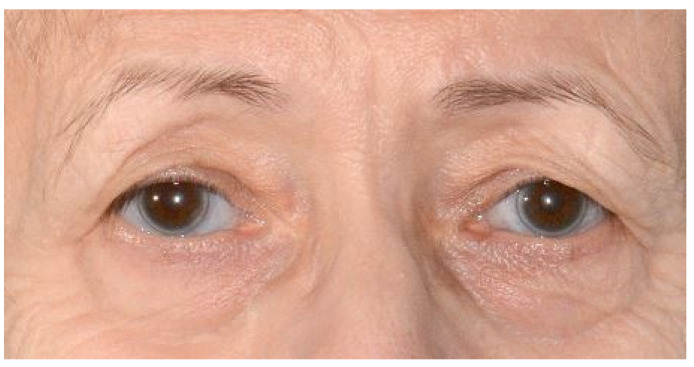
Characteristic bluish sclerae in a known case of osteogenesis imperfecta.

**Figure 3 ijerph-18-04169-f003:**
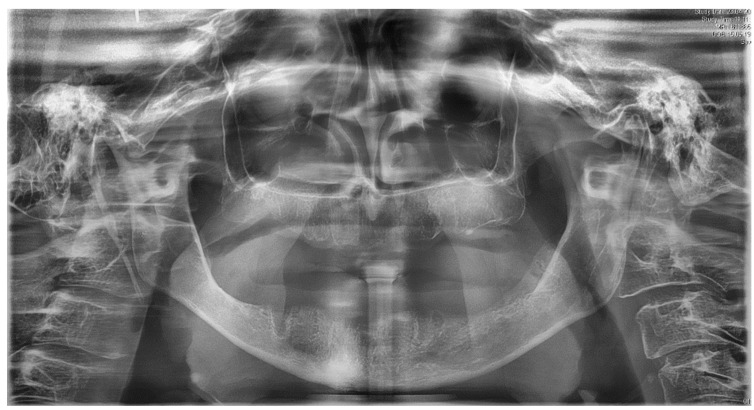
Preoperative panoramic view.

**Figure 4 ijerph-18-04169-f004:**
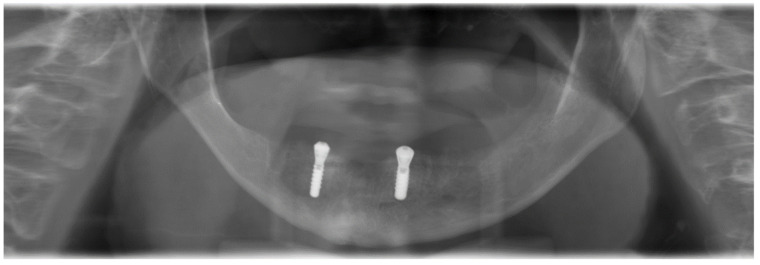
Postoperative panoramic view after the insertion of two interforaminal tissue-level implants.

**Figure 5 ijerph-18-04169-f005:**
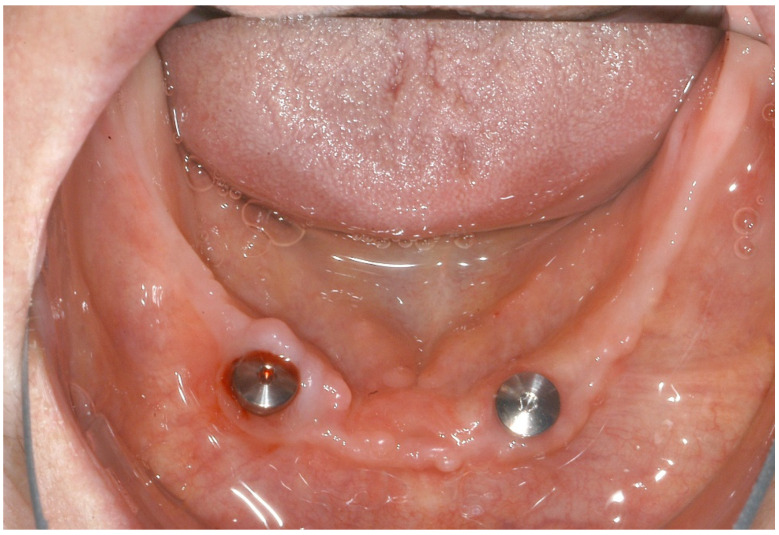
Healing caps after uncovery.

**Figure 6 ijerph-18-04169-f006:**
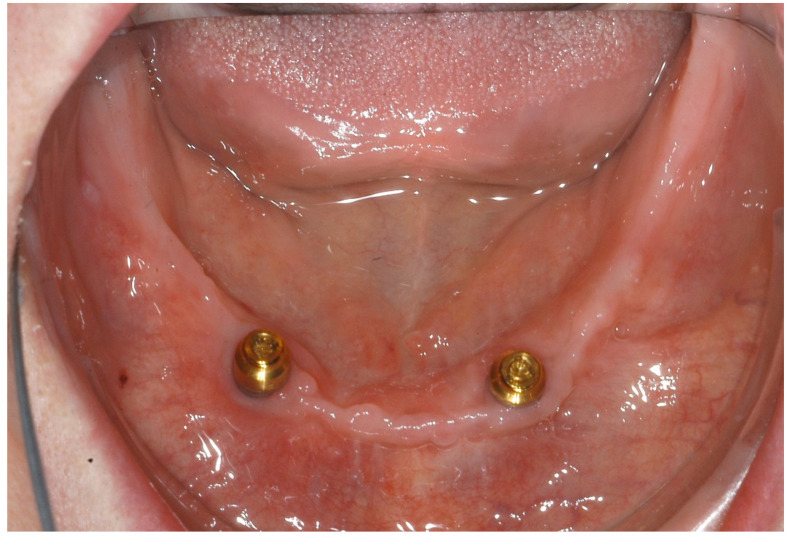
Prosthetic restoration with locators.

**Figure 7 ijerph-18-04169-f007:**
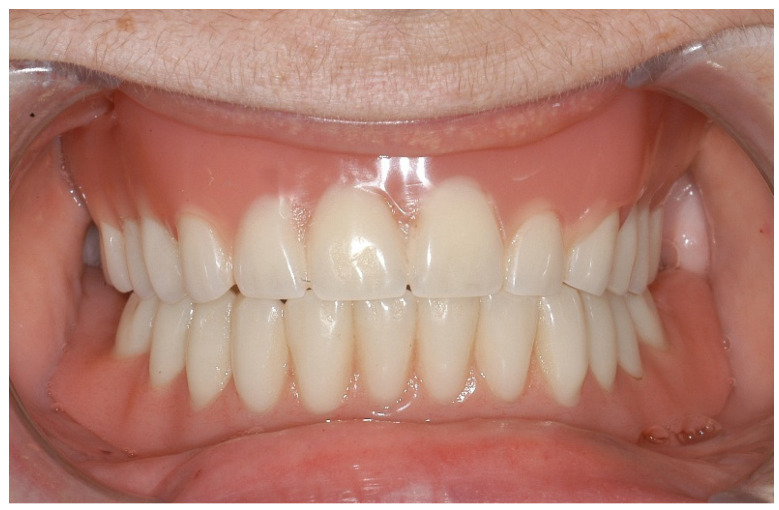
Final restauration after one year. The upper jaw was restored with a full denture.

**Figure 8 ijerph-18-04169-f008:**
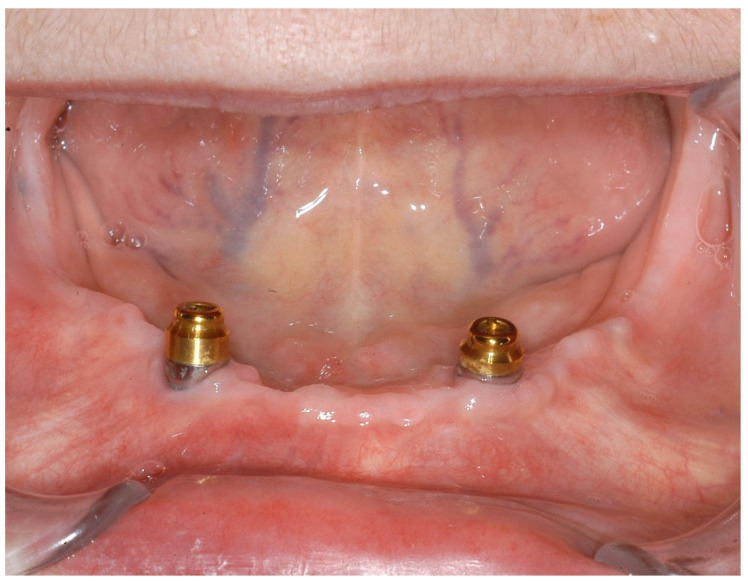
Clinical situation after one year.

## Data Availability

The datasets supporting the conclusions of this article are available from the Department of Cranio-Maxillofacial Surgery, University Hospital Münster, Germany.

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
