# Peer review of "Implant-Prosthetic Restoration of a Patient with Osteogenesis Imperfecta: A Case Report"

_ijerph, 2021, doi:10.3390/ijerph18084169_

Round 1
Reviewer 1 Report
To the Authors:
The manuscript "Implant-prosthetic restoration of a patient with osteogenesis imperfecta: a case report" addresses a topic of high clinical relevance, and it would benefit professionals and patients.
Considering the open access aspect of the Journal, it is expected that several professionals (specialists or not) would be able to read the paper. As osteogenesis accompanied by dentinogenesis is not routinely diagnosed and treated, the more information and details about the clinical case are provided, the greater the benefit to professionals and patients. Thus, some modifications are suggested aiming at the manuscript's improvement.
Abstract:
Line 10: Please replace the term "collagen I" with "type I collagen" throughout the text, starting from the Abstract.
Introduction:
Lines 29-30: providing some pathognomonic signs and symptoms of osteogenesis and dentinogenesis would help clinicians become aware of the alterations.
Case report:
Lines 45-46: What was the overall health status of the patient? For example, the pre-surgical condition and results from the exams.
Lines 52-54: was this condition documented? If yes, the pictures (of the tooth and the pair of dentures) could be inserted as part of this clinical case report.
Lines 58-60: the whole condition of soft tissues should be included.
Line 70: before starting the surgical procedure description, the treatment planning should be explored and detailed. Radiographs, models, and computer tomography should be included.
The diagnosis and treatment planning is crucial for the long-term success of the implants and prosthesis. Therefore, it is highly recommended that the authors provide information concerning both phases.
Lines 66-69: Has the patient signed a consent term before the treatment started? Did she authorize the publication of her case? This issue must be mentioned.
Lines 70-71: Please, write the prescription of clindamycin in full. Was the patient indeed intolerant to penicillin?
Line 79: Please, show the postoperative images documenting the "unremarkable" healing.
Line 81: Insert the 3-month follow-up image.
Line 91: Before starting the discussion, the images from the finalized clinical case are missing. How was the occlusion of the patient? If there is a 1-year follow-up, please provide the images.
Discussion and conclusions
The occlusion plays a vital role in the performance of the implants. Also, periodontal maintenance, hygiene, and diet control are essential for the treatment's long-term durability. Therefore, the authors should also address those issues. Although osteointegration is critical in the case of osteogenesis, the other related clinical factors are also relevant. Moreover, the objective of this report was "The aim of this article is therefore to report on the implant-prosthetic restoration of a patient with OI type I." The prosthetic consideration is missing.
Author Response
We would like to thank the reviewers for their time spent on reviewing our manuscript and their helpful comments. Their suggestions have been implemented in the manuscript. In this letter, we respond point-by-point to the comments and explain the revisions.
All changes to the manuscript were highlighted using the "Track Changes" function in Microsoft Word.
We hope the manuscript is now suitable for publication in the International Journal of Environmental Research and Public Health.
Rev 1:
To the Authors:
The manuscript "Implant-prosthetic restoration of a patient with osteogenesis imperfecta: a case report" addresses a topic of high clinical relevance, and it would benefit professionals and patients.
Considering the open access aspect of the Journal, it is expected that several professionals (specialists or not) would be able to read the paper. As osteogenesis accompanied by dentinogenesis is not routinely diagnosed and treated, the more information and details about the clinical case are provided, the greater the benefit to professionals and patients. Thus, some modifications are suggested aiming at the manuscript's improvement.
Abstract:
Line 10: Please replace the term "collagen I" with "type I collagen" throughout the text, starting from the Abstract.
Answer: we have now added "Collagen type I" throughout the manuscript.
Introduction:
Lines 29-30: providing some pathognomonic signs and symptoms of osteogenesis and dentinogenesis would help clinicians become aware of the alterations.
Answer: we added this paragraph: “Orofacial manifestations often associated with OI include DI, posterior open bite (lateral open bite), class III dental and skeletal malocclusion, anterior and posterior crossbites and impacted teeth [6].” (line 32-34)
Case report:
Lines 45-46: What was the overall health status of the patient? For example, the pre-surgical condition and results from the exams.
Answer: the patient was otherwise healthy. We added this to the case report (line 51)
Lines 52-54: was this condition documented? If yes, the pictures (of the tooth and the pair of dentures) could be inserted as part of this clinical case report.
Answer: the teeth were removed alio loco therefore we do not have any pictures. We added this information to the manuscript. (line 58)
Lines 58-60: the whole condition of soft tissues should be included.
Answer: The soft tissues were closed and inconspicuous, there was no swelling or redness. We added this information. (line 64-65)
Line 70: before starting the surgical procedure description, the treatment planning should be explored and detailed. Radiographs, models, and computer tomography should be included.
The diagnosis and treatment planning is crucial for the long-term success of the implants and prosthesis. Therefore, it is highly recommended that the authors provide information concerning both phases.
Answer: we have included the preoperative x-ray (Figure 3). We did not have a model for planning, nor did we have a 3-d x-ray as the patient was financially limited. The implants were placed clinically in the only two regions where bone was present.
Lines 66-69: Has the patient signed a consent term before the treatment started? Did she authorize the publication of her case? This issue must be mentioned.
Answer: we added this informations to the manuscript:
Institutional Review Board Statement: The study was conducted according to the guidelines of the Declaration of Helsinki, and approved by the Ethics Commission of Westphalia-Lippe and the Westphalia Wilhelms University Münster (2021-231-f-N).
Informed Consent Statement: Informed consent was obtained from the patient to publish this case report.
Lines 70-71: Please, write the prescription of clindamycin in full. Was the patient indeed intolerant to penicillin?
Answer: Yes, the patient was intolerant to penicillin. We added this information. (line 82)
Line 79: Please, show the postoperative images documenting the "unremarkable" healing.
Answer: Unfortunately, we did not take any photos postoperatively. However, we have taken photos of the uncovery of the implants and have now included them (Figure 5).
Line 81: Insert the 3-month follow-up image.
Answer: unfortunately we have no pictures of the follow-up after 3 months
Line 91: Before starting the discussion, the images from the finalized clinical case are missing. How was the occlusion of the patient? If there is a 1-year follow-up, please provide the images.
Answer: We have included the clinical photos with the final restoration and those of the implants after one year (figures 7,8).
Discussion and conclusions
The occlusion plays a vital role in the performance of the implants. Also, periodontal maintenance, hygiene, and diet control are essential for the treatment's long-term durability. Therefore, the authors should also address those issues. Although osteointegration is critical in the case of osteogenesis, the other related clinical factors are also relevant. Moreover, the objective of this report was "The aim of this article is therefore to report on the implant-prosthetic restoration of a patient with OI type I." The prosthetic consideration is missing
Answer: we have included figures 7 and 8 to show the prosthetic restoration. These figures also show the very good oral hygiene. The upper jaw was restored with a full denture. We have included this information. (line 109-110)
Rev2:
Dear Authors, the article is well written and very interesting from the clinical point of view: the possibility to treat patients affected by osteogenesis imperfecta is very helpful for the community and people with healthy disparities. However, before acceptance, your paper need some improvements:
- do you have an ethical approval to publish the case report?
Answer: we added this informations to the manuscript:
Institutional Review Board Statement: The study was conducted according to the guidelines of the Declaration of Helsinki, and approved by the Ethics Commission of Westphalia-Lippe and the Westphalia Wilhelms University Münster (2021-231-f-N).
Informed Consent Statement: Informed consent was obtained from the patient to publish this case report.
2) you should improve introduction and discussion, correlating your results with the ones obtained in immunocompromised patients
3) you should improve introduction and discussion, hypothesizing (as you cite in the abstract) the use of biomaterials - bone substitutes in such patients.
To explain how these procedures (the same described in the case report) can be useful and applicable also to immunocompromised patients with a good follow-up (PubMed ID: 26238779, 26238779)
To explain how biomaterials ca be useful to reconstruct bone also in such patients, like the described case report (PubMed ID: 21841997)
how implants can be placed in osteoporotic patients (PubMed ID: 33644012)
how implants can be placed in patients affected by sjogren syndrome (PubMed ID: 33412784)
how implant can be placed in systemically compromised patients (PubMed ID: 33213704)
how biomaterials can improve the bone volume of the jaws prior to implant placement, in particular in patients like the described case report. (PubMed ID: 33238625)
Answer: many thanks for the valuable tips! we have incorporated the recommended literature into the manuscript. (line 46-48; line 151-153)
Reviewer 2 Report
Dear Authors, the article is well written and very interesting from the clinical point of view: the possibility to treat patients affected by osteogenesis imperfecta is very helpful for the community and people with healthy disparities. However, before acceptance, your paper need some improvements:
1) do you have an ethical approval to publish the case report?
2) you should improve introduction and discussion, correlating your results with the ones obtained in immunocompromised patients
3) you should improve introduction and discussion, hypothesizing (as you cite in the abstract) the use of biomaterials - bone substitutes in such patients.
To explain how these procedures (the same described in the case report) can be useful and applicable also to immunocompromised patients with a good follow-up (PubMed ID: 26238779, 26238779)
To explain how biomaterials ca be useful to reconstruct bone also in such patients, like the described case report (PubMed ID: 21841997)
how implants can be placed in osteoporotic patients (PubMed ID: 33644012)
how implants can be placed in patients affected by sjogren syndrome (PubMed ID: 33412784)
how implant can be placed in systemically compromised patients (PubMed ID: 33213704)
how biomaterials can improve the bone volume of the jaws prior to implant placement, in particular in patients like the described case report. (PubMed ID: 33238625)
Author Response

(The authors gave the same response as above.)
